# Selective excitation enables encoding and measurement of multiple diffusion parameters in a single experiment

Neil MacKinnon[1], Mehrdad Alinaghian[1], Pedro Silva[1,2], Thomas Gloge[3], Burkhard Luy[3], Mazin Jouda[1], and Jan G. Korvink[1]

[1]Institute of Microstructure Technology (IMT), Karlsruhe Institute of Technology, Hermann-von-Helmholtz-Platz 1, D-76344 Eggenstein-Leopoldshafen, Germany.
[2]DeepSpin GmbH, Kurfürstenstraße 56, 10785 Berlin
[3]Institute of Biological Interfaces (IBG-4), Karlsruhe Institute of Technology, Hermann-von-Helmholtz-Platz 1, D-76344 Eggenstein-Leopoldshafen, Germany.

**Correspondence:** Neil MacKinnon (neil.mackinnon@kit.edu); Jan G. Korvink (jan.korvink@kit.edu)

**Abstract.** Band selectivity to address specific resonances in a spectrum enables one to encode individual settings for diffusion experiments. In a single experiment, this could include different gradient strengths (enabling coverage of a larger range of diffusion constants), different diffusion delays, or different gradient directions (enabling anisotropic diffusion measurement). In this report a selective variant of the bipolar pulsed gradient, eddy-current delay (BPP-LED) experiment enabling selective encoding of three resonances was implemented. As proof-of-principle, the diffusion encoding gradient amplitude was assigned a range dependent on the selected signal, thereby allowing the extraction of the diffusion coefficient for water and a tripeptide (Met-Ala-Ser) with optimal settings in a single experiment.

## 1 Introduction

There is little dispute over the importance of diffusion as a physical effect, and its influence on many natural processes.

Diffusion poses a limiting factor in many industrial processes, such as the mixing of chemical reagents to achieve a specific product with sufficient yield. Understanding the interplay between intrinsic diffusion rates and operational time dependent parameters need to be properly understood and carefully considered in designing such manufacturing processes. Nuclear magnetic resonance (NMR) has proven to be uniquely capable to extract diffusion constants for a specific chemical substance and, in favorable cases, even mixtures, and also to study and understand the processes of diffusion at all length scales and within all compartments available to the molecule under consideration. Thus NMR has facilitated experimental proofs of Fick's laws, and has facilitated precise measurements of anisotropic effects arising from spatially extended molecules in solution, and subtracting the effects due to geometry, or ionic charges, and the like.

In magnetic resonance imaging, diffusion is currently the limiting spatial resolution factor for inductive NMR detection, because excited spins in a molecule tend to diffuse away from their excitation site whilst waiting for pulse readout. Whereas this uncertainty cannot be removed completely, shorter pulse sequences have the effect of improving resolution by reducing the distance a spin ensemble can wander.

Spatial resolution is especially important in such areas as brain science and brain diagnostics, where anisotropic diffusion tensor imaging is the primary noninvasive means to discover the sub-cellular structure of a specific brain (Göbel-Guéniot et al. (2020)). Essentially, diffusing spin ensembles do not readily traverse cell walls and axons, so that the local structure of brain tissue renders an anisotropic response, with directional weighting being in favour of gliding along the cell walls. In this way, diffusion has helped to reveal brain connectivity patterns. But also in studies of nanoporous materials, anisotropic diffusion parameters have been used to reveal local nanostructure.

Diffusion anisotropic measurements aim to reveal the 3-dimensional pattern of molecular movement. Mathematically, the movement at a spatial point has to be resolved in three orthogonal directions, which typically would require at least three motionally sensitive measurements, each in combination with a spatial gradient aligned with the specific direction of measurement. The most time-consuming part of the pulse sequence is the echo time, and the recovery time. Spatial encoding will additionally require the three measurements for each spatial voxel, further slowing down the acquisition.

A degree of freedom available to spectroscopic diffusion measurements is the spectral dispersion, taking advantage of the individual resonances as a means to encode additional information within a single experiment. This is particularly interesting for molecular mixtures whose components vary in the physical dimensions and thus their diffusion coefficient, or for a tracer molecule in an anisotropic environment where diffusion is dependent on direction. Instead of performing multiple diffusion measurements, each optimized for a particular regime, a frequency bandwidth could be selectively encoded with appropriate diffusion parameters independent of another, different bandwidth, itself encoded with differing diffusion parameters. This could be extended down to encoding individual resonances in a spectrum with diffusion parameters appropriate for extracting diffusive properties with high precision.

There are two additional benefits to integrating selective elements to the experiment: first is the elimination of dominating, uninteresting signals (typically solvent) to improve sensitivity to minor components. This has been demonstrated for protein diffusion in water using a selective version of the stimulated echo experiment (Yao et al. (2014)) and a selective version of the spin echo experiment for measuring minor components in a mixture (Howe (2017)). The second benefit, noted by both Yao and co-workers (Yao et al. (2014)) and Howe (Howe (2017)), is to reduce spectral congestion and thereby improve DOSY data analysis, an active field of research in extracting diffusion coefficients from overlapping signals (Aguilar et al. (2010); Colbourne et al. (2011); Foroozandeh et al. (2018); Lin et al. (2020)).

In this report, we have extended the stimulated echo experiment using bipolar gradients and longitudinal eddy current compensation (BPP-LED) (Wu et al. (1995)) to enable selective diffusion encoding. As proof-of-concept we demonstrate the ability to selectively address up to three individual resonances (each with a bandwidth of $60\,\mathrm{Hz}$), each encoded with independent diffusion parameters.

In this paper we are **specifically paying tribute** to Geoffrey Bodenhausen, a pioneer of NMR spectroscopy, on the occasion of his seventieth birthday. Our chief source of inspiration comes from one of his earliest papers, published in 1976 (Bodenhausen et al. (1976)) whilst he was in Oxford, and which explores selective excitation, as well as numerous of his papers that have explored the measurement of diffusion by various ingenious means.

## 2  Materials and Methods

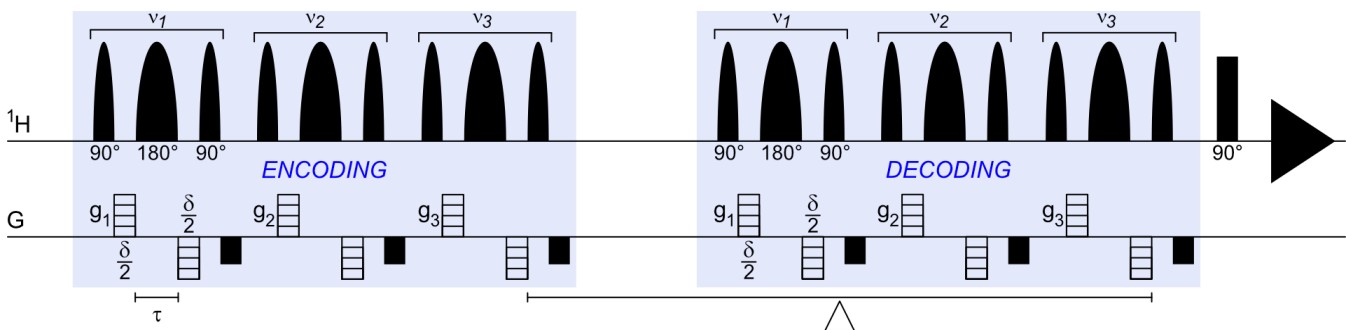

**Figure 1.** The selective BPP-LED (bipolar pulse paired, longitudinal eddy-current delay) diffusion experiment: Sel-BPP-LED. In this work, three spectral regions with center frequency $\nu_n$ were encoded. For each selective spectral region, $g_n$ were defined separately for $\nu_1$ (water) and $\nu_{2,3}$ (MAS peptide, Met and Ala methyl signals). Despite a hard $90°$ read pulse, using the same phase cycle as the BPP-LED sequence, the resulting spectrum consists of only the selected spectral regions (Figure 2). While not demonstrated in this work, diffusion encoding for each selected frequency could be uniquely defined beyond using $g_n$ by including $\delta_n$, inserting additional delays between the $\nu_n$ decoding segments to vary $\Delta_n$, and the direction of $g$ (where hardware permits). All parameters used this work are described in the Materials and Methods..

*Materials.* Deuterium oxide ($D_2O$) and the tri-peptide Met-Ala-Ser (MAS) were purchased from Sigma Aldrich and used as received. All $H_2O$ was of MQ quality. An NMR sample in a $5\,\text{mm}$ NMR tube was prepared by dissolving $10\,\text{mg}$ MAS in $500\,\mu\text{L}$ of 90:10 $D_2O{:}H_2O$.

*NMR Spectroscopy.* NMR experiments were performed using an Avance III 500 MHz wide-bore NMR spectrometer (Bruker Biospin, Ettlingen) using a Micro5 micro-imaging probe equipped with a $5\,\text{mm}$ NMR detector. Sample temperature was controlled using water cooling flowing through the imaging gradient sleeve, maintained at $20\,°C$.

Non-selective diffusion experiments were performed using the bipolar gradient, longitudinal eddy current compensated experiment (BPP-LED) (Wu et al. (1995)) (Bruker pulse program ledbpgp2s1d). The diffusion-encoding gradient pulse was the smoothed square (SMSQ10.100) with length $2\,\text{ms}$, gradient recovery time was $200\,\mu\text{s}$, gradient strength was varied using the parameter optimization 'popt' function in TopSpin 3.6.3 (Bruker BioSpin) from 1 to 95% of maximum gradient strength through 16 experiments, and the diffusion time was $250\,\text{ms}$.

Selective diffusion experiments were performed using a modified version of the non-selective experiment (Sel-BPP-LED) - $90°$ hard pulses were replaced by sinc shaped pulses, and $180°$ refocusing pulses were replaced by Gaussian refocusing pulses

(Sinc1.1000 and Gaus1_180r.1000 from the Bruker shaped pulse library), Figure 1. The selective pulse length and power were calculated using the Shape Tool (TopSpin 3.6.3) using a selective bandwidth of 60 Hz, resulting in excitation pulses of 26.8 ms and refocusing pulses of 14.7 ms. The diffusion time was $\Delta = 250$ ms, with the timing in the pulse program modified taking into account the additional selective blocks in the pulse sequence. Three signals were chosen for the selective diffusion experiment: the water signal (4.8 ppm), and from the MAS peptide the methyl signal of methionine (2.1 ppm) and the methyl signal of alanine (1.45 ppm). As in the non-selective experiment, the popt function was used to vary the maximum gradient amplitude over 16 experiments. The simultaneous optimization feature of the popt function was used to control individually the three gradient amplitudes, $g_n$ (Figure 1).

All NMR experiments were performed using RF pulses calibrated on the day of measurement. Each measurement consisted of 8 scans, each scan containing 32 K data points over a spectral width of 20 ppm. The FIDs were zero-filled by a factor of 2 and multiplied by an exponential function equivalent to 0.3 Hz line width prior to Fourier transformation. The series of 16 spectra from each diffusion measurement was extracted using an in-house Matlab script (Matlab 2020b, Mathworks, Natick, USA). The Matlab 'fit' function was used to extract the diffusion coefficients. We note that the dosy functionality available in TopSpin was *not* used in setting up the experiments nor in the data analyses.

Diffusion coefficients were extracted by fitting the intensity versus applied gradient strength using the equation (Wu et al. (1995); Sinnaeve (2012)):

$$I = exp(-\gamma^2 G^2 \delta^2 D(\Delta - \delta/3 - \tau/2)) \tag{1}$$

where $I$ is the normalized NMR signal intensity, $\gamma$ is the [1]H gyromagnetic ratio, $\delta$ is the diffusion gradient pulse length, $\Delta$ is the diffusion time, $\tau$ is the time between the bipolar gradient pulses, and $G$ is the applied gradient strength. It is noted that the value of $\tau$ becomes significant in the selective experiment (14.9 ms in this work), and that the equation would need to be specified separately for each selective frequency in case different selective pulse bandwidths, and therefore different refocusing pulse lengths, are to be used. There was no correction applied for using the smoothed square gradient shape which, should be noted, deviates slightly from an ideal rectangular pulse (Sinnaeve (2012)). The diffusion of water in the $D_2O:H_2O$ mixture was used to calibrate an apparent gradient strength assuming i) a diffusion coefficient D = $2.02 \times 10^{-9}$ m$^2$/s (taken for pure $H_2O$, Tofts et al. (2000); Holz et al. (2000)) and ii) rectangular gradients. The apparent gradient strength was determined to be $260\,\mathrm{G\,cm^{-1}}$.

## 2.1 Results and Discussion

A series of experimental NMR spectra measured using both BPP-LED and Sel-BPP-LED are presented in Figure 2, with a summary of details provided in Table 1. Comparing the the SNR when using the BPP-LED vs. Sel-BPP-LED experiments, the water signal decreased by 33%, while the MAS methyl signals of methionine and alanine were essentially unchanged (-4%, +2%, respectively). The Sel-BPP-LED experiment yielded intensity decay curves faithfully reproducing the non-selective experiment (Figure 3a). A slight variation in the extracted diffusion coefficients was observed (3.3%, 3.9%, and -1.5% relative deviation for $D_{water}$, $D_{MAS,Met}$ and $D_{MAS,Ala}$). The likely source for the deviations are selective RF pulse imperfections

| Signal | Selective (y/n) | SNR ($\times 10^3$) | Diffusion Coefficient ($\times 10^{-10}\ m^2\ s^{-1}$) |
|---|---|---|---|
| Water | n | $2700 \pm 700$ | $20.0 \pm 0.6$ |
| Water | y | $1800 \pm 130$ | $19.3 \pm 0.07$ |
| Met | n | $100 \pm 30$ | $4.33 \pm 0.03$ |
| Met | y | $102 \pm 8$ | $4.17 \pm 0.01$ |
| Ala | n | $90 \pm 20$ | $4.14 \pm 0.02$ |
| Ala | y | $86 \pm 6$ | $4.20 \pm 0.01$ |

**Table 1.** Non-selective (BPP-LED) and selective (Sel-BPP-LED) diffusion experiments are compared. The water (4.8 ppm), methionine methyl signal of MAS (2.0 ppm), and alanine methyl signal of MAS (1.4 ppm) were used for these analyses. SNR was calculated using the first increment in the diffusion series. Diffusion coefficients were extracted from the diffusion plots using Equation 1. The diffusion time was the same for all experiments ($\Delta = 250$ ms). A single Sel-BPP-LED experiment was used, with the water and MAS signals exposed to gradient strengths ranging from 1 - 60% and 1 - 95% of maximum, respectively. Reported SNR values are the mean $\pm$ standard deviation of triplicate measurements.

which could be compensated by using selective pulse shapes with better performance, for example Gaussian pulse cascades Q3 and Q5 (Emsley and Bodenhausen (1992)) or optimal control derived selective pulses (Matson et al. (2009)). In the case of all three signals exposed to 1 - 95% of gradient maximum, the extracted water diffusion coefficient was $(19.2 \pm 0.2) \times 10^{-10}$ $m^2\ s^{-1}$, a value with larger fit error (1.1 vs. 0.37%) and slightly larger deviation relative to the non-selective experiment (4.0 vs. 3.3%) compared to using a gradient strength spanning 1 - 60%. The extracted values for the MAS Met and Ala methyl signals were essentially unchanged ($(4.16 \pm 0.01) \times 10^{-10}\ m^2\ s^{-1}$ and $(4.20 \pm 0.01) \times 10^{-10}\ m^2\ s^{-1}$).

The increased degree of experimental freedom enabled by Sel-BPP-LED is demonstrated in Figure 2b,c and Figure 3b. Figure 2b reveals that the water signal fully decays well before the MAS when using the same gradient amplitude across all signals. Adjusting the gradient amplitude range for the water signal to apply 1 - 60% of maximum strength was found to be sufficient to reach full decay at the end of the experiment. In the *same* experiment, the MAS signals were simultaneously exposed to gradient amplitudes ranging from 1 - 95% of maximum strength, appropriate to sample the intensity decay curve for the larger molecule. As a result, the Sel-BPP-LED experiment enables parallel diffusion measurements of molecules of vastly different diffusive properties by selectively encoding the appropriate diffusion experimental parameter (i.e. gradient amplitude in this case) into a well resolved signal.

Selective variants of numerous NMR experiments have been realized since the idea of converting multi-dimensional to 1D experiments using selective pulses was reported (Kessler et al. (1986)). Such experiments are often exploited to simplify otherwise complex spectra (Kiraly et al. (2021); Alexandersson et al. (2020); Poggetto et al. (2017); MacKinnon et al. (2016); Pelupessy et al. (1999)), to suppress strong signals that dominate the dynamic range (Cutting et al. (2000)), or to selectively drive desired coherence pathways (Haller et al. (2019); Ferrage et al. (2004); Chiarparin et al. (1998)). Diffusion experiments have also benefited from the addition of selectivity, to reduce the influence of strong solvent signals in biomolecular samples

(Shukla and Dorai (2011); Yao et al. (2014)), to focus the diffusion experiment onto select molecules in a complex mixture (Lyu et al. (2018)), or to use the diffusion dimension as an access point to further selective-based NMR experiments (e.g. selective TOCSY (Lyu et al. (2018)) and supplemented with relaxation encoding (Poggetto et al. (2017))).

The selective experiment described in this report offers the same advantages in terms of signal suppression and reducing complexity. A key difference compared to previous selective diffusion reports is the encoding of information into individual resonances. This is in contrast to the method described by Howe (Howe (2017)), who demonstrated diffusion measurements by selective excitation of four resonances. Since every resonance was exposed to the same experiment parameters, all four resonances could be simultaneously excited resulting in a net gain in time per scan compared to the experiment described here. Yao and colleagues demonstrated a selective diffusion experiment using selective pulses of larger bandwidth ($\sim 2\,\mathrm{kHz}$) (Yao et al. (2014)) and therefore also gained in experimental time per scan. The benefit of the Sel-BPP-LED experiment (in fact we are not limited to BPP-LED, any stimulated echo based-diffusion experiment should be compatible with the concept) is the ability to parallelize measurements, accessing diffusion information from multiple sample species in a single experiment.

The ability to accelerate measurements via parallelization will be important in many applications. For example, considerable advantages for increased measurement time efficiency for DTI (Diffusion Tensor Imaging) applications are anticipated, since the 3D diffusion tensor (with up to 6 components) needs to be determined on a voxel-by-voxel basis. In this case, the ability to encode up to 6 scalar components in a single k-space acquisition, i.e. in one shot and subsequent FID, would be a tremendous advantage. For technical systems, this could be achieved by using fluids of designed composition such that at least 6 well resolved signals are available. For clinical DTI, the measurement will be reliant on identifying molecules that are sufficiently abundance and chemical shift resolution at typical MRI scanner field strengths (e.g. at 3 T candidates could include lactate, N-acetyl amino acid derivatives, choline derivatives, creatine derivatives (Wilson et al. (2019))); however, even if all six tensor components cannot be encoded there will still be an measurement acceleration when more than one component can be encoded. An additional example is complex samples containing both slow and fast diffusing species, where restricted diffusion is present. Using a selective diffusion experiment, the optimal parameters can be simultaneously used for the fast diffusing species and, in the example presented in Figure 4, the restricted slow-diffusing species.

The limit to the number of selective units, $n$, in the selective diffusion experiment is governed by i) the desired bandwidth to be selectively encoded, ii) the desired diffusion time $\Delta$, and, related, iii) the $T_1$ relaxation of the selected signals. For the worst case scenario, selectivity with small bandwidth of multiple resonances with short $T_1$ will not be possible since the resulting diffusion time would be too long and signal intensity would be lost to relaxation. In the experiment demonstrated here, a selective bandwidth of $60\,\mathrm{Hz}$ was used exclusively, requiring selective pulse lengths of $26.8\,\mathrm{ms}$ and $14.7\,\mathrm{ms}$ for excitation and refocusing. This limited the diffusion times $\Delta > 3 \times 68.3\,\mathrm{ms}$ for the 3 signals selectively encoded, or more generally $\Delta > n \times (2 \times t_{90,sel} + t_{180,sel} + \delta)$. Shorter diffusion times can be accessed by selectively encoding spectral regions instead of individual resonances, thereby reducing the required length of the selective pulse (in the case of J-coupled signals within the excited bandwidth, there will be minor signal loss to anti-phase coherence which will be destroyed by the gradient applied after magnetization storage). This has been demonstrated for protein and peptide diffusion measurements, selectively exciting bandwidths on the order of $2\,\mathrm{kHz}$ with the benefit of relatively short selective pulse lengths on the order of $2.5\,\mathrm{ms}$ (Yao et al.

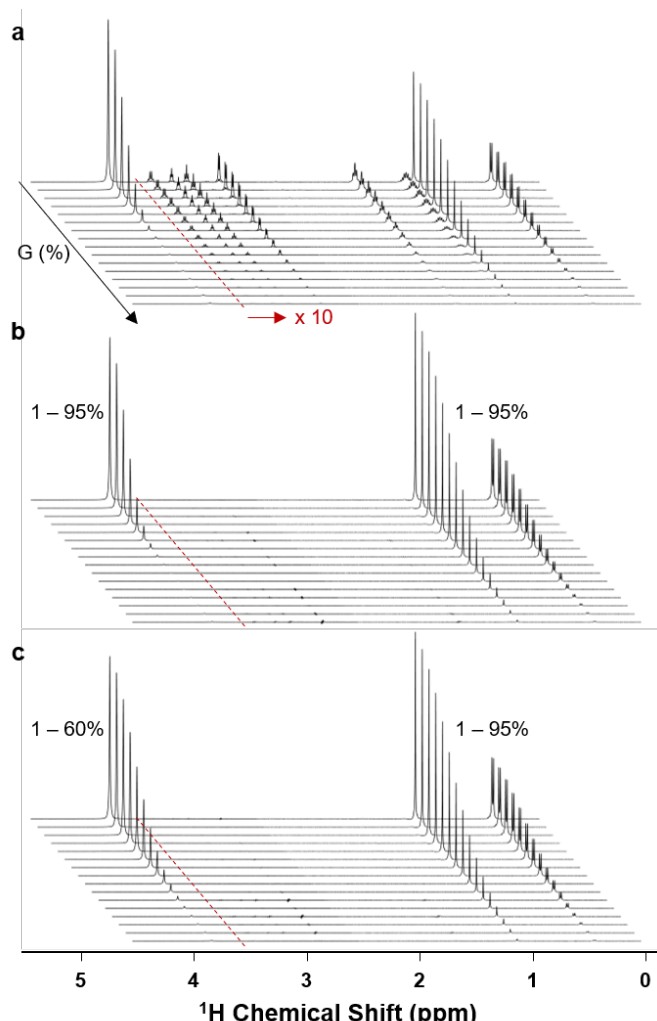

**Figure 2.** Exemplary $^1$H NMR spectra of $20\,\mathrm{mg\,mL^{-1}}$ Met-Ala-Ser (MAS) tripeptide, non-selective (a) and selective (b,c) diffusion experiments. The resonances used for selective encoding were: water (4.8 ppm), Met methyl group of MAS (2.1 ppm), and Ala methyl group of MAS (1.5 ppm). Demonstrated in c, using selective encoding the gradient amplitude experienced by the water signal could be controlled independent of that experienced by the MAS signals (1-60% vs. 1-95%). In all plots, the intensity of the region from 1 - 4.5 ppm was increased by a factor of 10 for clarity (highlighted in a).

(2014)). Encoding more than one resonance with the same diffusion parameters would also be possible using pulses with multi-band selectivity, effectively reducing $n$ and also enabling access to shorter diffusion times.

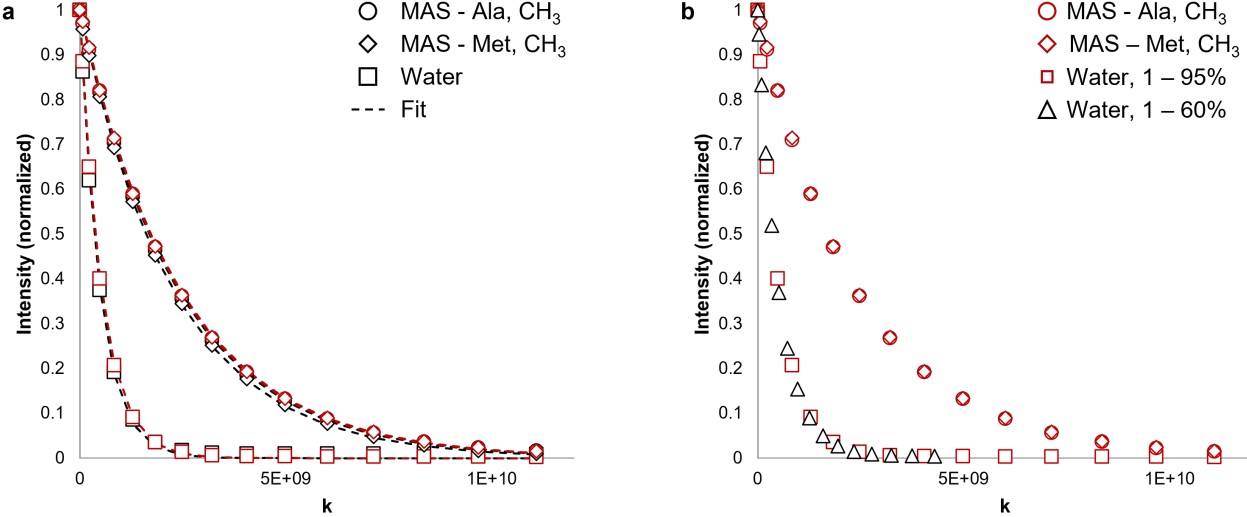

**Figure 3.** a) Comparison of signal intensity versus the diffusion experiment parameter $k = \gamma^2 G^2 \delta^2 (\Delta - (\delta/3) - (\tau/2))$ for the alanine and methionine methyl signal of Met-Ala-Ser (MAS, circles, diamonds) and the water signal (squares). Black symbols: the experiment was non-selective; red symbols: selective excitation was used. The dashed lines are the fits to the data using Equation 1. The adjusted $R^2$ values were all $> 0.999$. The diffusion time was $\Delta = 250\,\mathrm{ms}$. b) Comparison of signal intensity versus $k = \gamma^2 G^2 \delta^2 (\Delta - (\delta/3) - (\tau/2))$ for the water (squares, diamonds) and MAS-Ala, Met methyl (circles, diamonds) signals using Sel-BPP-LED. Using the selective experiment, the applied gradient strength experienced by the water signal could be varied independently of the applied gradient strength experienced by MAS. Diamonds: applied gradient strength varied from 1 - 60%. Using the selective excitation experiment, MAS signals were always exposed to an applied gradient varying from 1 - 95%. The experiments were performed in triplicate, the mean values are plotted. The diffusion time was $\Delta = 250\,\mathrm{ms}$.

## 3    Conclusions

Taking advantage of individual resonances as a means to encode additional, unique information into a single experiment is a means to achieve measurement parallelization. For each unique encoding $n$, experimental time is accelerated by a factor $n$. In this work a Sel-BPP-LED diffusion experiment demonstrated this concept, with $n = 2$ in this case since two signals belonged to the same molecule and were encoded with the same gradient direction. While demonstrated for the BPP-LED experiment, the concept is general and can be applied to any diffusion pulse sequence implementing the stimulated echo. Selective encoding enables the simultaneous measurement of diffusion coefficients of sample mixture components experiencing different transport properties dictated by molecular size or selective restriction via matrix interactions. This principle could be extended to anisotropic diffusion by encoding gradient directions into different resonances. To improve the experiment, selective RF pulses with better performance should be explored to avoid systematic deviations in the extracted diffusion coefficients.

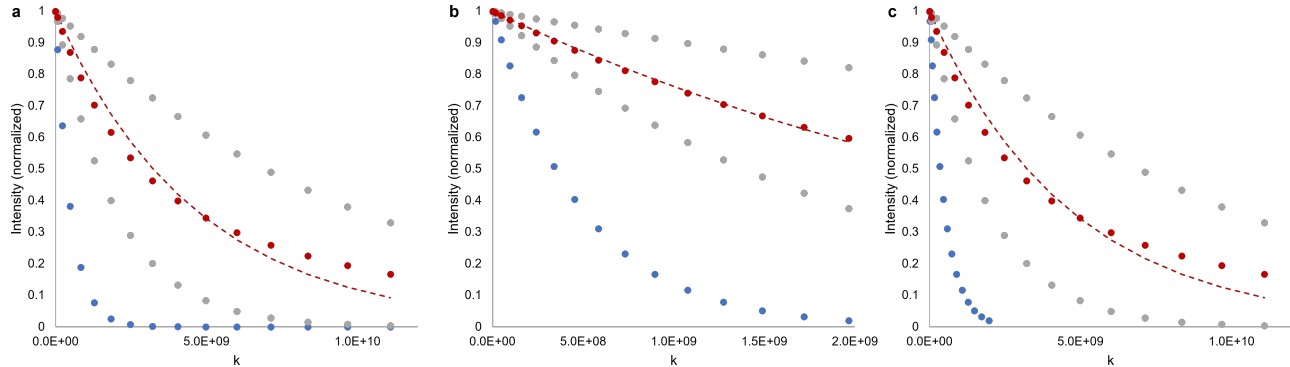

**Figure 4.** Simulated data to demonstrate the distinction between a fast-diffusing mixture component, and a slow-diffusing mixture component experiencing restriction. a) Gradient strength sufficient to observe multi-component decay (failure of single-exponential fit, red), but not optimized for fast component (blue). b) Gradient strength optimized for fast diffusing species (blue), but not sufficient to identify multi-component diffusion (red). c) Single experiment, gradient strength optimized independently for fast and multi-component species. Dashed lines are fits to the diffusion equation (Equation 1) assuming single exponential decay – the red dots are the sum (50:50) of two single exponential decays (gray). Simulated $D = 2 \times 10^{-9} \ m^2 \ s^{-1}$ (blue), $5 \times 10^{-10} \ m^2 \ s^{-1}$, and $1 \times 10^{-10} \ m^2 \ s^{-1}$ (gray), $G = 260.8 \ \mathrm{G \, cm^{-1}}$, $\delta = 2 \, \mathrm{ms}$, $\Delta = 250 \, \mathrm{ms}$, $\tau = 0.21 \, \mathrm{ms}$.

*Code availability.* The Sel-BPP-LED pulse program, tested using TopSpin 3.6.3, is available at https://doi.org/10.5281/zenodo.5105713.

*Data availability.* The NMR data is available at https://doi.org/10.5281/zenodo.5105713.

*Author contributions.* Conceptualization: PS, MJ, JGK. Formal Analysis: NM. Investigation: NM, MA, TG. Supervision: BL, MJ, JGK. Visualization: NM. Writing - original draft: JGK and NM. Writing - review and editing: all authors.

*Competing interests.* The authors declare no competing interests.

*Acknowledgements.* J.G.K. acknowledges the support from an EU2020 FET grant (TiSuMR, 737043), ERC-SyG (HiSCORE, 951459), the DFG under grant KO 1883/39-1 optiMUM, the framework of the German Excellence Intitiative under grant EXC 2082 "3D Matter Made to Order", and together with B.L. the KIT-VirtMat initiative "Virtual Materials Design II". J.G.K., M.J., T.G., B.L., and N.M. acknowledge the partial financial support of the Helmholtz Association through the programmes "Science and Technology of Nanosystems – STN", "BioInterfaces in Technology and Medicine – BIFTM" and "Materials Systems Engineering – MSE". The authors acknowledge the support of the Karlsruhe Nano Micro Facility (KNMFi), a Helmholtz Research Infrastructure at Karlsruhe Institute of Technology.

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
