# Peer review of "Selective excitation enables encoding and measurement of multiple diffusion parameters in a single experiment"

_Magnetic Resonance, 2021_

## Author Response (AR1)

We sincerely thank the community and reviewers for their time and critical feedback to our manuscript. We believe we have addressed all of the comments, and the manuscript has gained significantly in clarity. Below are our point-by-point responses, together with the adjustment that was made in the manuscript:

Reviewer 1 (Davy Sinnaeve):

The work by MacKinnon et al. describes a new concept for diffusion experiments, where multiple selective excitation/refocusing blocks are used not only to excite different segments of the spectrum, but also encode diffusion in a different way for each segment in a single scan. This is demonstrated here by using different gradient strengths for the diffusion encoding of the water and the MAS peptide signals, but indeed could be expanded upon towards other parameters such as gradient direction. This concept is useful, and one can indeed think of ways to apply this to other diffusion pulses sequences. It is certainly of interest for the readership of MR.

The authors mention that wider segments of the spectrum can also be selected, encompassing multiple signals. But what if the signals within one segment are coupled to one other? Would that not generate artefacts from non-refocused J-coupling evolution?

If several spins are selected by the selective pulses that are coupled to each other, coupling evolution indeed takes place during each selective encoding/decoding element. As a result, some in-phase signal will be transferred to antiphase, which will then be removed by the crusher gradient at the end of each selective element. As such, only a reduction in in-phase signal intensities will be experienced in the weak coupling limit. The signal intensities are reduced according to cos(pi J teff), where teff is given by the time of selected spins spent in the transverse plane, which depends on the gradient lengths, delays, and the specific selective pulses used in the sequence. For wider selected areas corresponding selective pulses are typically short, on the order of 1 or few milliseconds. As a consequence teff is also in the range of few ms and with typical small 1H-1H-coupling constants on the order of -15 to +10 Hz, effective signal reductions are on the order of 10% (assuming a coupling constant of 15 Hz and teff=8ms, the cos-dependence causes a 7% reduction in intensity). This can usually be tolerated.
If coupled spins experience severe second order artefacts, signal distortions might occur, but these distortions will be the same for each gradient strength, so that diffusion measurements should still be possible.

Manuscript adjustment: a statement noting the reduction in signal in case of J-coupling has been added to the discussion section.

The SNR for the Ala methyl signal increased significantly for the selective experiment compared to the non-selective experiment (Table 1). This is unexpected, given the expected increased relaxation losses during the selective pulses. Is there an explanation for this? Is the same observed for the Methionine methyl signal?

After a closer look at the SNR script, we identified an error in the noise calculation. This has been corrected, results and discussion updated.

Manuscript adjustment: Recalculated SNR values Ala, Met, Water = -2, +4, -33% difference between selective and non-selective experiments.

The SNR and diffusion coefficient measured on the Methionine methyl group are not shown in Table 1, not discussed in the text, and the decay curves are not shown in Figure 3. Yet this was the third signal selected. It would be good to include this, to observe the consistency with the Ala signal.

Figure 3 updated.

[Figure]

Updated figure, including 1. Diffusion data for the Met signal, 2. Changing the axis in (b) to k instead of Gradient Strength (for consistency).

Some remarks about the representation of the pulse sequence of Figure 1.

- It would be useful for the sake of clarity if the flip angles of the selective and hard pulses are indicated (90°, 180°). Also, it would be useful if it can be clearly indicated somewhere that the order of selective pulse frequencies in the encoding and decoding blocks are the same, as actually this is not strictly necessary (even though the effective âˆ † would vary then).
  Pulse sequence diagram has been updated.

- The way that the diffusion delay âˆ † is indicated in the pulse sequence is misleading, since now the wrong suggestion is made that it is the delay between the last selective pulse of the encoding block and the first selective pulse of the decoding block. The conventional definition of âˆ † actually also requires including either the full encoding block (from the beginning of the first gradient pulse of the encoding block until the beginning of the first gradient pulse of the decoding block), or the full decoding block (from the end of the last gradient pulse of the encoding block until the end of the last gradient pulse of the decoding block).
  Pulse sequence diagram has been updated.

- The pulse sequence also shows gradient pulse durations can be different for each selected frequency, i.e., $\delta_n/2$. In principle this is indeed possible, but not used here. In fact, looking at the Bruker pulse sequence code provided by the authors on zenodo, this appears hard coded to be the same for each block (p30). Here, it is in fact the gradient strength g that varies with n, and should be indicated as $g_n$ in the figure to make it consistent with the demonstration in this work.

Pulse sequence diagram has been updated.  Caption description has been updated to reflect the actual experiment demonstrated in this work, and to highlight which parameters could be varied.

- Finally, it would be nice to also explicitly indicate τ from the ST equation (or perhaps as τ$_n$, if different 180° pulse durations are envisaged).
Pulse sequence diagram has been updated. The new version is a better representation of what was actually used in this work, and the figure caption notes where the flexibility in the sequence lies.

Manuscript adjustment: new figure for pulse sequence.

[Figure]

Updated pulse sequence.

The Stejskal-Tanner equation (1) is valid for rectangular gradient shapes. The gradient shape is not mentioned in the experimental part. It would be good to confirm the gradient shape in the experimental part, since for Bruker these are standardly SINE or SMSQ shapes, requiring a slightly modified ST equation.

Manuscript adjustment: We used SMSQ gradients, and is noted in the materials and methods section. A clarification statement is added declaring that no correction for slightly non-ideal rectangular pulses was applied in this work.

Also, it should maybe be explicitly noted that the value of τ is actually quite significant here to correctly take into account, since it also encompasses the long selective 180° pulse (14.7 ms). If different selective pulse durations would be used for each spectral segment (as is possible in the pulse sequence code provided on zenodo), the ST-equation will be significantly different for each segment.

Manuscript adjustment: this clarification was added to the materials and methods section.

Figure 3a shows the signal intensity as a function of k, while Figure 3b shows it as a function of gradient strength percentage. This inconsistency seems unnecessary to me.

Manuscript adjustment: Figure was updated.

On page 3, line 66, ledbpge2s1d should be ledbpgp2s1d.

Manuscript adjustment: correction made.

Reviewer 2 (anonymous):

In this article, MacKinnon et al. describe a pulse sequence that makes it possible to use different diffusion-encoding parameters for different resonances in a single experiment. To achieve this, they replace all the hard pulses of a stimulated-echo sequence by selective ones, except for the final 90° pulse, and they introduce a loop for the dephasing and rephasing blocks. They illustrate the results on a sample consisting of a tri-peptide in H2O:D2O. The pulse sequence may be relevant for the analysis of samples that have components with widely different diffusion coefficients. In its present form, however, the manuscript does not provide sufficient information to show or explain the benefits of the new sequence. I recommend that it be reconsidered after major revisions.

The exact benefit of using different gradient ramps for the different resonances should be clarified. Is it expected to improve the precision of the measured diffusion coefficient? The trueness? Both? This is hard to understand in the present manuscript. Maybe the authors should report the uncertainty of the fit for the selective and non-selective experiments? Also, in Table 1, the value of the diffusion coefficient for water measured with the selective experiment is further way from the calibration value that than measured with the non-selective experiment. Please explain if that it an improvement, and why?

We observed small deviations in the extracted diffusion coefficients between the selective and non-selective experiment variants. While a reduction in the error was observed for the extracted water diffusion coefficient in the optimized selective experiment, the benefit was small (data added to the manuscript). We aim to emphasize that the benefit of the selective experiment is measurement parallelization. As an example, the technique is expected to bring considerable advantages in speedup for DTI (Diffusion Tensor Imaging) applications, since the 3D diffusion tensor (with up to 6 components) needs to be determined on a voxel-by-voxel basis. In this case, the ability to encode up to 6 scalar components in a single k-space acquisition, i.e. in one shot and subsequent FID, would be a tremendous advantage. Another example is a restricted diffusion situation, where one can simultaneously measure with optimal diffusion parameters a fast species and a slow species experiencing restriction. A plot of simulated data is presented to highlight this example.

Manuscript adjustment: a new paragraph was added to the discussion. The anticipated gain in DTI and multi-component diffusion is discussed. A new figure was added: 3 plots of simulated diffusion curves demonstrating that in a single experiment, one can identify slow, multi-component diffusion at the same time as a fast, single component diffusing species.

[Figure]

Distinction between a fast-diffusing mixture component, and a slow-diffusing mixture component experiencing restriction. a) Gradient strength sufficient to observe multi-component decay (red curve), but not optimized for fast component (blue). b) Gradient

strength optimized for fast diffusing species (blue), but not sufficient to identify multi-component diffusion (red). c) Single experiment, gradient strength optimized independently for fast and multi-component species.  Dashed lines are fits to the diffusion equation assuming single exponential decay – the red dots are the sum (50:50) of two single exponential decays (gray). Simulated D = 2e-9 m2 /s (blue), 5e-10 m2/s, and 1 e-10 (gray), G = 260.8 G/cm, delta = 2 ms, Delta = 250 ms, tau = 0.21 ms.

Why does SNR increase in the selective experiment for the MAS signal? Is that a positive observation? Or is that a reflection of the fact that coherence transfer pathway selection is not as effective?

After a closer look at the SNR script, we identified an error in the noise calculation.  This has been corrected, results and discussion updated.

Manuscript adjustment: Recalculated SNR values Ala, Met, Water = -2, +4, -33% difference between selective and non-selective experiments.

Please provide the values of the encoding gradient in G/cm or T/m, or the maximum gradient that can be delivered by the probe. These probes typically deliver up to 200 or 300 G/cm. Together with the diffusion delay of 250 ms and a ramp of up to 95%, this would results in an unusually strong attenuation. Please clarify.

The value of the gradient was determined to be 260 G / cm.  The diffusion experiment parameters were adjusted to observe the decay of MAS. With the same parameters, the water signal decayed with 40% of maximum gradient strength (Figure 3).

Manuscript adjustment: value of the gradient strength inserted into methods section.

The authors state that the experiment is accelerated by a factor of n. In the example that they report, n = 3 but two of the resonances belong to the same molecule, and are encoded with the same parameters. So, isn't the acceleration by a factor of 2 only?

This is a valid point, here we demonstrate n = 2 since the parameter we adjusted was the gradient amplitude, which was the same for the two MAS signals. The general expected acceleration is still true, so long as the n encodings are unique.

Manuscript adjustment: clarification was added to the conclusion.

More generally, stating that the experiment is accelerate will only be valid once the authors convincingly explain why there is an advantage of using different ramps.

Addressed in the related comments.

The author state that the approach is applicable to any diffusion experiment. This is not obvious, in particular for spin echo based experiment. Once one of the resonances is in the transverse plane, it experiences any subsequent gradient pulse. How would it work to encode independently the different resonances ?

This is indeed true, we had incorrectly over-generalized the applicability of this concept. The method requires magnetization storage along the z-axis, and therefore is more correctly applicable to stimulated echo methods. This statement has been clarified in the manuscript.

Manuscript adjustment: clarification that the experiment can be generalized to stimulated echo experiments was implemented in the discussion and conclusion.

It would help to discuss examples where one would want to use this pulse sequence. If the gradient ramp are tailored for the selected resonances, it means that the corresponding diffusion coefficients are know at least approximately. So what would the accelerated experiment be used for? The introduction covers a broad range of applications of diffusion NMR, but no connection is made, in the discussion, between these area of application and the proposed method. For example, the abstract states the possibility to encode multiple delay or multiple directions; does that assume that several resonances are available for the same molecule?

The discussion has been updated to highlight the applicability (also in response to an earlier comment), in particular in connection with the concepts from the introduction.

Manuscript adjustment: new paragraph was added to the discussion.

Two minor points:

It would be helpful to also show the pulse sequence without the loop structure, that is, the complete pulse sequence with a fixed value of n. Please also indicate, in this figure, the exact definition of Delta. This would help the reader to see that the Delta delay is the same for all of the selected resonances.

The pulse sequence diagram has been updated accounting for all reviewer suggestions.

Manuscript adjustment: Figure 1 updated.

Why use two different steps to encode the two MAS resonances, rather than a single one using a dual-band selective pulse?

We agree that using multi-band pulses can be used in this experiment, which was originally argued as a means to effectively access shorter diffusion times since less time is spent in selective encoding (last sentence of the discussion section). In this work, it was our object to demonstrate the 3 signals could be selectively treated, even if 2 of the signals originated from the same molecule.

Manuscript adjustment: the last sentence of the discussion was updated, exchanging "multi-selectivity" with "multi-band selectivity" to clarify this point.

Community comment (R. Soong)

This is an excellent and well written paper for the NMR community regarding the use of NMR for diffusion measurement.

In the past, the mesurements of molecular diffusion require mulitple experiments with different parameter for optimization.  In this case, the Aurhtors use a single measure to measure mulitple parameter, providing a significant time saving.

 Here is a couple of questions

1) Have the author try to use this measurment in liquid crystal enviroment to evaluate anisotropic diffusion tensors?

We do have preliminary data of a selective diffusion experiment in a PBLG sample.  In the experiment, three signals originating from a single molecule free to diffuse within the aligned sample were exposed to X, Y, and Z diffusion encoding / decoding gradients selectively, in a single experiment (i.e. measurement of diffusion in 3 directions in one experiment).  Our preliminary data suggest there was an observable difference in the diffusion behavior between the Z and X,Y diffusion directions.  We require significantly more time and experiments to confirm this result, and hope to report in the near future.

2) How realistlic to use this sequence to extract diffusion tensor and map out the orientation of its environment.

We believe this could be one of the key experiments standing to gain in terms of measurement time from the selective diffusion experiment. In principle, 6 separate measurements are required to define the diffusion tensor, and in practice many more directions are often measured. By increasing the number of directions measured in a single experiment by taking advantage of the selective encoding, the measurement time required to define the tensor should significantly improve.

---

## Author Response (AR2)

Karlsruhe Institute of Technology

**Institute of Microstructure Technology**

Executive Director: Prof. Dr. J. G. Korvink

Hermann-von-Helmholtz-Platz 1
76344 Eggenstein-Leopoldshafen, Germany

Phone: +49 721 608-29314
Fax: +49 721 608-24331
Email: neil.mackinnon@kit.edu
Web: www.imt.kit.edu

Official in charge: Dr. Neil MacKinnon
Our reference: FuE2-nm-9026-21
Date: 2021-11-07
Pages: 1

KIT-Campus North | IMT | P.O.Box 3640 | 76021 Karlsruhe, Germany

Dr. Fabien Ferrage
Magnetic Resonance

[Figure]

[Figure]

**Correction to manuscript submission**

Dear Dr. Ferrage,

We thank you and the reviewers for their input to our manuscript. We have modified the manuscript taking into account the comments of the Reviewer. The highlighted manuscript is below: please see the Materials and Methods section.

Please feel free to contact me if any further corrections / additions are required.

Yours sincerely and on behalf of all co-authors,

Karlsruhe Institute of Technology
Institute of Microstructure Technology

Dr. Neil MacKinnon

Karlsruhe Institute of Technology (KIT)
Kaiserstr. 12
76131 Karlsruhe, Germany
USt-IdNr. DE266749428

President: Prof. Dr.-Ing. Holger Hanselka
Vice Presidents: Michael Ganß, Prof. Dr. Thomas Hirth,
Prof. Dr. Oliver Kraft, Christine von Vangerow,
Prof. Dr. Alexander Wanner

LBBW/BW Bank
IBAN: DE18 6005 0101 7495 5012 96
BIC/SWIFT: SOLADEST600

[revised manuscript text omitted]